# **Countervailing regional snowfall patterns dampen Antarctic surface mass variability**

Jeremy Fyke<sup>1</sup>, Jan Lenaerts<sup>2,3</sup>, and Hailong Wang<sup>4</sup>

<sup>1</sup>Los Alamos National Laboratory

<sup>2</sup>Institute for Marine and Atmospheric Research, Utrecht University, Utrecht, Netherlands

<sup>3</sup>Department of Atmospheric and Oceanic Sciences, University of Colorado, Boulder, USA

<sup>4</sup>Pacific Northwest National Laboratory, Richland, USA

Correspondence to: Jeremy Fyke (fyke@lanl.gov)

Abstract. Snowfall over Antarctica, the dominant term of the Antarctic surface mass balance, displays large regional heterogeneity in temporal variability patterns. This heterogeneity has the potential to dampen variability in integrated Antarctic surface mass trends by counteracting increases in snowfall in one location with decreases in another (and vice versa). To examine the presence of countervailing regional snowfall patterns, here we present an analysis of spatial patterns of regional

- Antarctic snowfall variability, their broader climate drivers and their impact on integrated Antarctic snowfall variability simulated as part of a preindustrial 1800 year equilibrated Earth System Model simulation. Correlation and composite analyses based on this output allow for a statistically robust exploration of Antarctic snowfall variability. We uncover statistically significant countervailing snowfall patterns across Antarctica that are corroborated by regional modelling and ice core records. These countervailing patterns are driven by variability in large-scale atmospheric moisture transport and cause large spatial
- heterogeneity in temporal variability, with a dampening effect on overall Antarctic snowfall variability magnitude. This dampening has implications for regulation of Antarctic-sourced sea level variability, detection of an emergent anthropogenic signal in Antarctic mass trends and identification of AIS mass loss accelerations.

#### 1 Introduction

Precipitation of snow over Antarctica is the means by which the Antarctic Ice Sheet (AIS), currently the largest contiguous
ice complex on the planet, accumulates mass. Over millennia, this accumulation has resulted in the sequestration of 57m of eustatic sea level equivalent as AIS ice (Fretwell et al., 2013). As a result of the large area of this capacitor of freshwater, natural variability in AIS-integrated snowfall has a leading impact on AIS mass trends, which in turn influence global mean sea level (GMSL) variability.

AIS-integrated accumulation variability is the net effect of regional accumulation variability signals. Previous studies have

20 leveraged Gravity Recovery and Climate Experiment (GRACE) output, satellite altimetry and/or regional modeling to highlight strongly heterogeneous spatial patterns of temporal variability (Mémin et al., 2015; Martín-Español et al., 2016; Shepherd et al., 2012; Boening et al., 2012). Complementing these remotely-sensed observations, longer pointwise time series of accumulation variability from ice cores (.e.g. Frezzotti et al., 2013; Thomas et al., 2017, and references therein) also suggest the presence

of heterogeneous regional patterns of snowfall variability. Historical climate reanalysis (Dee et al., 2011) as well as regional models driven by this reanalysis (e.g. Van Wessem et al., 2014) further confirm the presence of high spatial heterogeneity in recent historical Antarctic accumulation variability trends (Wang et al., 2016).

- These analyses provide critical insight into historical drivers of ice sheet mass balance variability and hint at a potentially 5 strong role for heterogenous patterns of snowfall variability in dampening AIS-integrated snowfall and associated ice sheet mass variability. They are also important in understanding future change, during which regional variability will be superimposed on by a strong secular trend towards greater accumulation (Palerme et al., 2016; Frieler et al., 2015; Previdi and Polvani, 2016; Lenaerts et al., 2016) and increased low-elevation surface melting (Fyke et al., 2010; Trusel et al., 2015). However, they are characteristically hampered by either short time series length, sparse point data, observational uncertainty, a mixture of
- forced and natural variability signals, or a combination of all of these. As a result, a clear analysis of spatial heterogeneity in Antarctic snowfall variability - and the impact of this heterogeneity on AIS-integrated snowfall variability and associated mass change - is currently lacking.

Analysis of global climate model output provides a potential pathway around these difficulties. In particular, pre-industrial climate model control simulations - long quasi-equilibrium simulations in which no time-varying external forcing is applied

- (e.g. Eyring et al., 2016) provide spatially complete, temporally extensive representations of natural climate variability that are uncontaminated by external forcing changes. Motivated by the potential for climate model output to provide robust dynamical insight into drivers of spatially heterogeneous Antarctic accumulation variability, here we use the Community Earth System Model (Hurrell et al., 2013) to explore spatial patterns in Antarctic snowfall variability. The study is presented as follows: in Section 2 we describe the model, simulation and analysis methods we adopt to explore Antarctic accumulation variability. In
- Section 3 we detail the findings of our analyses, which are further interpreted in Sections 4 and 5.

#### 2 Methods

We explore AIS basin-scale accumulation variability and its link to broader climate patterns using output from the Community Earth System Model (CESM, version 1). CESM is a comprehensive coupled Earth system model that is used for a diverse range of applications, including recent explorations of Southern Hemisphere climate (Holland et al., 2016; Lenaerts et al.,

- 2016). The particular CESM simulation we focus our analysis on is a ~1 degree resolution preindustrial control simulation of the CESM Large Ensemble experiment (CESM LE hereafter) (Kay et al., 2015). Output from this fully coupled simulation provides a robustly equilibrated (and thus statistically stationary) representation of global climate system dynamics that is uncontaminated with anthropogenic or other external climate forcings, and of sufficient length (1800 years) to support robust variability analysis. These benefits provide an important advantage of this approach, over analogous studies based on reanalysis
- data from the relatively short and non-stationary recent historical period. Our methodological approach is similar to other investigations of unforced Southern Hemisphere climate variability, that also use output from climate model preindustrial control simulations (e.g. Santoso et al., 2006; Holland et al., 2005; Previdi and Polvani, 2016).

In our analysis, we represent regional Antarctic accumulation heterogeneity by disaggregating the AIS into individual AIS drainage basins (Zwally et al., 2012) (Figure 1). Ice shelves are included in this basin disaggregation. This approach is adopted because each basin represents a distinct dynamical integrator of regional climate signals, making a basin-scale analysis a practical and glaciologically tractable method for linking climate-driven snowfall variability to regional Antarctic glaciological

- trends. At the basin scale, we focus primarily on drivers of variability in annually-averaged, spatially integrated precipitation rates (units of Gt/yr) because of the importance of this measure for mass balance and dynamics, combined with the relative insensitivity of long-term AIS ice sheet dynamics to higher-frequency variability (e.g. related to the seasonal cycle and/or individual storm events, which nonetheless are important in summing to the net annual signal). We apply several analysis tools to quantify simulated basin-scale snowfall variability, compare inter-basin variability relationships, and link basin-scale variabil-
- ity to broader patterns of climate variability. We first perform correlation analysis between basin-specific annually-averaged accumulation time series to identify inter-basin accumulation variability relationships. In addition, for each basin, we construct composite climatologies based on CESM-simulated annually-averaged accumulation to construct paired monthly-resolved characteristic climate states associated with high/low basin accumulation years. This allows examination of the relationships between basin accumulation timeseries and broader modes of climate variability. To verify that the patterns of variability
- identified in CESM model output were robust, we repeated subsets of our analysis on output from a 27 kilometer resolution RACMO2.3 regional atmosphere/ice sheet surface model simulation. RACMO2.3 has been extensively applied and validated over Antarctica, thus providing an important benchmark of CESM performance (Lenaerts et al., 2016). Finally, we also also assess the similarity of CESM results against compilations of accumulation variability derived from ice core records.

#### **3** Results

## 20 3.1 Simulated preindustrial Antarctic accumulation variability

CESM1.1-simulated AIS accumulation over the historical period has been analyzed in detail in Lenaerts et al. (2016). Given similar model provenance, the spatial pattern of accumulation in the CESM LE control simulation is similar to that validated by Lenaerts et al. (2016), with lowest values in the dry interior and high values around the coast, particularly along the Amundsen Sea coastline and the Antarctic Peninsula (Figure 2a). The model produces a climatological integrated precipitation of 2168

- 25 Gt/yr, of which ~0.1% falls as liquid (as such, for the purposes of this study we consider the terms 'precipitation', 'snowfall' and 'accumulation' equivalent, given that the absolute value and variability of sublimation is small relative to precipitation). This value is lower than recent historical mean CESM/RACMO2.3 accumulations assessed in Lenaerts et al. (2016) and Van Wessem et al. (2014) (2428/2829 Gt/yr), due to the significant positive trend in historical CESM-simulated integrated Antarctic accumulation combined with an overall negative CESM snowfall bias.
- 30 The spatial pattern of interannual variability in modeled accumulation, measured as the standard deviation of the climatological annual-averaged accumulation, largely mirrors the spatial pattern of climatological accumulation magnitude: where the magnitude is high, so too is the variability (Figure 2b). Accumulation variability is highest around the Antarctic coast, and reaches maximum values between Marie Byrd land and the Antarctic Peninsula. In contrast, minimum variability occurs over

the East Antarctic plateau. The standard deviation of AIS-integrated annually-average accumulation variability is 98 Gt/yr, which is lower than recent historical variability in both CESM and RACMO2.3 (122/135 Gt/yr), consistent with lower overall climatological accumulation. Ranking of basin-scale accumulation contributions to overall variability highlights the leading influences of basin 20 (Marie Byrd Land of West Antarctica) and, secondarily, basins 12-14 (Adelie Land to Wilhelm II Land of East Antarctica) in regulating overall variability due to very high basin-integrated accumulation rates despite relatively small

size (for the former) and large extent despite relatively low accumulation rates (for the latter).

Relative variability in Antarctic accumulation can be assessed via the coefficient of variation (CV), which is calculated as the standard deviation divided by the mean (e.g. Turner et al., 1999, Figure 12). The spatial pattern of CESM-simulated CV (Figure 2c) differs notably from the spatial patterns of both mean accumulation and accumulation variability, indicating an

- influence of both in regulating relative variability. Low site-specific CV values arise over both the Antarctic Peninsula (AP) and coastal WAIS and along major AIS ice divides. This pattern is also apparent in RACMO2.3 output (not shown). High mean accumulation drives low CV over the AP and coastal WAIS, while low CV along ice divides is likely related to low variability at these locations, perhaps associated with the large influence of steady background clear-sky precipitation (Bromwich, 1988). Conversely, relatively high variability combined with relatively low mean accumulation conspire to generate highest CV values
- over the Ross and Amery ice shelves, Victoria and George 5 Lands, Dronning Maud Land and the eastern Ronne-Filchner ice shelf. This pattern is also apparent in RACMO2.3 (van de Berg et al., 2006) and also reflected in earlier estimates of Antarctic accumulation variability (Turner et al., 1999).

AIS-integrated CV calculated using the mean and standard deviation of AIS-integrated accumulation equals 0.04. Reassuringly, this value is very similar to the recent historical AIS-integrated CV generated by both CESM and RACMO2.3 (0.06/0.04).

- However, it is very low relative to CESM-simulated CV values at individual grid points on the model domain (average of 0.2). This large difference between AIS-integrated and local values also holds when comparing AIS-integrated CV to CV calculated at the basin scale, which is also higher as evidenced qualitatively by the wider distributions of normalized basin-scale accumulation relative to the AIS-wide integrated distribution (Figure 2d) and quantitatively by a mean CV of 0.17 across basins - more than 4 times higher than the ice-sheet-integrated value. This much larger basin-averaged CV value, relative to
- the value calculated from the AIS-integrated snowfall time series, implies the presence of countervailing snowfall patterns that strongly dampen the contribution of regionally high regional accumulation variability to the AIS-integrated variability signal. It also suggests that the impact of regional AIS accumulation variability is quickly obscured, in cases where aggregation to larger spatial scales is employed. On the other hand, the similarity of basin-based and pointwise-based CV averages (0.17/0.2) indicates that little information about variability is lost in basin-scale aggregation, thereby supporting our use of basin-scale
- disaggregation in exploration of the interplay between regional ice sheet variability patterns.

### 3.2 Inter-basin countervailing accumulation signals

To identify the spatial patterns of regional accumulation variability that dampen AIS-wide integrated snowfall variability, we examine zero-lag temporal correlations in CESM-simulated time series of annual-average integrated accumulation between all drainage basins. Figure 3a shows the matrix of resulting correlation coefficients, while Figure 4 shows the spatial distribution

of inter-basin accumulation correlations for a select set of basins chosen simply to represent 4 quadrants of Antarctica (plots for all basins available in Supplementary Material).

The presence of not only dampening but actively counteracting basin-specific integrated accumulation signals is demonstrated by the presence of multiple statistically significant (95% confidence interval) inter-basin negative correlations with

- respect to basin-scale integrated accumulation (blue squares in Figure 3a). On average, while each basin's accumulation variability is significantly (r>95%) positively correlated with 9 other basins (of 11 total average positive correlations per basin), each basin is also significantly *anti-correlated* with 12 other basins (of 16 total average negative correlations per basin). The presence of significantly anti-correlated inter-basin accumulation variability patterns reflects the fact that high/low snowfall years in one basin are synonymous with low/high snowfall years in other basins, with directly cancelling effects on the in-
- tegrated variability signal. This cancellation highlights spatially widespread and coherent 'actively-cancelling' patterns of Antarctic snowfall variability, the absence of which would tend to reduce the matrices of Figure 3 to insignificant correlations at increasing distances from the diagonal.

The spatial patterns of countervailing variability associated with the matrix of correlation coefficients in Figure 3 are highly basin-specific, indicating that inter-basin correlation patterns depend highly on the basin of interest. Figure 4a-d highlights

- the large diversity of spatial correlation patterns for an set of ice sheet drainage basins simply chosen to represent the four quadrants of Antarctica. Analagous figures for all basins can be found in the Supplementary Material. Accumulation in basin 2 demonstrates positive correlations with accumulations in nearby basins, mirrored by negative correlations with more remote basins. Conversely, accumulation in basin 7 correlates positively and significantly with accumulation in both neighbouring and remote basins, as well as correlating negatively and significantly with a broad and coherent region of the interior. Basin
- 14 shows only weak positive correlation levels to neighbouring basins and, largely, no significant correlation to more distant basins. Finally, the pattern of correlations based on basin 19 shows a prominent correlation dipole, with high/low accumulation in basin 19 and surrounding basins being countered by low/high accumulations in easterly WAIS basins and the Antarctic Peninsula.

## 3.3 Comparison to RACMO2.3 and ice core basin-scale variability

- Are the patterns of countervailing Antarctic snowfall simulated by CESM realistic? We approach this question by first comparing CESM accumulation variability patterns with that of the well-evaluated 1979-2015 RACMO2 Antarctic accumulation product (Van Wessem et al., 2014). Figure 3b presents the matrix of RACMO2 inter-basin annual accumulation correlations (equivalent to the analogous CESM matrix, Figure 3a). RACMO2 snowfall time series at each point are detrended prior to aggregation to the basin scale and correlation. As with CESM, RACMO2 simulates the presence of both multiple positive
- and, more importantly, negative inter-basin correlations. The presence of the latter robustly supports the CESM-based finding of dampening of the ice sheet-wide accumulation variability by countervailing regional variability patterns. While RACMO2 generates higher absolute positive/negative inter-basin correlation magnitudes (average 13/14 per basin) they are less statistically significant correlations than in CESM. This likely arises from the short (37 year) RACMO2 time series length relative to

the CESM time series length (1800 years) and highlights the need for sufficient record length for the purposes of statistically robustly characterizing ice sheet-related internal variability (e.g. Wouters et al., 2013).

Ice and firn cores provide an additional opportunity to directly assess CESM-simulated accumulation variability. A recent compilation of AIS ice core-derived snowfall variability records was compiled by the Past Global Changes (PAGES) Antarctica

- 2k working group (Thomas et al., 2017). As part of this work, Thomas et al. (2017) assessed local multi-ice-core composite 5 records in seven sub-regions of Antarctica against other sub-regions and RACMO2 and ERA-Interim output. Despite significant noise inherent to annual-scale analysis of Antarctic ice core data, this analysis also suggested the presence of countervailing variability signals, most clearly arising between the Antarctic Peninsula region and the western WAIS and between East and West Antarctica. Given the technique of utilizing recent-historical RACMO2.3 and ERA-Interim data to link far-field accu-
- mulation correlations to local accumulation variability observed in ice cores in Thomas et al. (2017), statistical confidence is 10 limited by short RACMO2.3/ERA-Interim time series length - again reflecting the need for longer analyses. Yet despite this drawback, the emergence of countervailing variability patterns in ice core data qualitatively supports similar findings in CESM.

#### 3.4 **Regional controls on countervailing Antarctic accumulation variability**

Having established the presence of strong countervailing patterns of Antarctic snowfall variability, we next analyze the un-15 derlying climatic drivers. We use basin-scale compositing, where for each ice sheet drainage basin global, monthly-resolved composite climatologies associated with low (5th percentile) and high (95th percentile) basin-scale accumulation years are constructed (n $\sim$ =90 for each composite). Differencing these basin-based composite climatologies highlights changes in regional climate conditions associated with basin-specific snowfall variability.

- Because precipitation over Antarctic ice drainage basins is largely dependent on advection of moisture from remote sources as opposed to local recycling (Sodemann and Stohl, 2009; Singh et al., 2016) we first assess basin-specific composite differ-20 ences in vertically integrated atmospheric zonal and meridional moisture fluxes in the expectation that basin-scale variability will be reflected in local atmospheric moisture convergence changes. Indeed, for all basins a consistent change in proximal offshore atmospheric moisture transport anomalies emerges with high accumulation years co-occurring with positive local onshore moisture transport anomalies and low accumulation years co-occurring with weaker on-shore transport anomalies (Figure
- 25 5 and Supplemental Material). Within each basin, the difference in absolute moisture flux tends to weaken inland, reflecting the dominance of coastal orographically-driven precipitation in Antarctica (Favier et al., 2013). Given that basin-integrated accumulation mostly reflects coastal precipitation, the compositing technique preferentially clusters years with greater coastal precipitation, leading to a tendency towards composite differences that are coastally maximized. Moisture flux anomalies are often largest in the local vicinity of the basin for which the compositing was based, identifying local circulation variability as an important final link between remote evaporative source regions and final precipitation location.
- 30

The tight relationship between Antarctic basin-scale accumulation and proximal moisture transport variability confirms a leading role for moisture transport in determining local basin snowfall variability. However, at a broader scale, the composite climatologies also identify spatially coherent large scale moisture transport changes as the causal factor behind interbasin countervailing accumulation variability. Specifically, for all basins that demonstrate simultaneous positive/negative accumula-

tion correlations with the basin for which compositing procedure is based on, the compositing also identifies corresponding onshore/offshore moisture transport anomalies. As an arbitrary demonstrative example, each basin with snowfall variability that is positively correlated with basin 7 (Figure 4b) also demonstrates increased onshore moisture transport, with the strength of the correlation tending to increase with increased onshore moisture transport changes. Such basins include several proximal

- basins, as well as remote basins on the AP and in Victoria land. The converse relationship holds for anti-correlated basins that tend to occur as a swath of basins across the interior of the EAIS. In general, this relationship between basin accumulation correlations and local and remote moisture transport changes holds across all basin-specific composites, strongly implicating large-scale modes of polar atmospheric moisture advection as the active driver of countervailing basin-scale accumulation variability.
- Moisture flux anomalies associated with basin-specific accumulation variability, as well as broader patterns of interbasin dampening/countervailing accumulation variability, are tightly linked to composited differences in both 500 hPa geopotential height (z500, contours in Figure 5) and sea level pressure (SLP, not shown). Stemming from the similarity between SLP and z500 composite differences (r=0.91) we limit our attention to the relationship of moisture flux changes to z500 variability. Basin-based accumulation compositing reveals spatially extensive and significant z500 differences with a transition from posi-
- tive to negative anomalies indicating large changes in horizontal pressure gradients typically lying in close proximity to the basin from which compositing was based (see, for example, Figure 5a, b and d). Moisture flux anomalies, which are character-istically maximal close to the composited basin, are consistently ~90 degrees leftwards of the maximum z500 gradient. This points to regional geostrophic atmospheric variability as a main determinant of the moisture transport variability that in turn drives inter-annual basin-scale accumulation variability.
- Despite the apparently robust control of atmospheric circulation variability on Antarctic basin-scale snowfall variability, another potential regulator of atmospheric moisture transport to Antarctica must also be considered, namely, regional sea ice concentration (SIC)-regulated local evaporative source region variability (Tsukernik and Lynch, 2013; Thomas and Bracegirdle, 2015). Compositing reveals that basin-scale accumulation variability is typically associated with nearby offshore SIC: basinspecific high-accumulation years tend to be characterized by statistically significant local decreases in upwind SIC, while low-
- accumulation years are characterized by upwind SIC increases (Figure 6). The broad co-occurrence of low/high upwind SIC with downstream high/low basin-scale accumulation suggests a role for sea ice-regulated local evaporation source variability in determining basin-scale accumulation variability. However, we argue against this causal relationship, because strong increases in P-E proximal to the basin upon which compositing was performed are *not* associated with opposing regions of negative P-E change over proximal sea ice loss regions (Figure 6). Instead, P-E decreases over these regions are typically weak and in some
- cases even positive (e.g. Figure 6a/b). This contradicts the pattern that would be expected if SIC-regulated evaporative increases were a dominant factor in locally increased Antarctic basin-scale snowfall: namely, a strong proximal offshore P-E decrease (reflecting increased evaporation that is not countered by opposing over-ocean precipitation) closely associated with the pattern of negative SIC change next to basins experiencing increased snowfall. The argument for a weak dependence of AIS snowfall on local SIC variability also holds for the case of basins experiencing countervailing snowfall but with reversed tendencies
- of change. Thus, we conclude that regional sea ice variability is not the primary factor driving basin-scale Antarctic snowfall

variability. Rather, both sea ice and Antarctic snowfall variability are both commonly regulated by atmospheric circulation variability, which leads to the appearance of similar (but not causal) spatial variability patterns. This conclusion contradicts speculations by Tsukernik and Lynch (2013); Thomas and Bracegirdle (2015); Lenaerts et al. (2016) but supports the findings of Singh et al. (2016) and recent studies exploring sea ice controls on Greenland Ice Sheet climate (Noël et al., 2014; Stroeve et al., 2017).

#### 3.5 Links to broader patterns of atmospheric variability

Establishing broad-scale atmospheric circulation variability as the main determinant of basin-scale snowfall variability, and broader patterns of inter-basin variability dampening and cancellation, subsequently allows for a broader assessment of the role of larger-scale atmospheric circulation variability in controlling these features. Notably, the pattern of atmospheric circulation differences extracted via compositing of basin-scale snowfall clearly reconstructs with remarkable fidelity previously-

characterized broad-scale patterns of variability, highlighting their important impact.

The CESM-simulated Amundsen-Bellinghausen Sea Low (ABSL) appears as perhaps the most dominant control on individual basin-scale accumulation variability and associated regionally countervailing accumulation patterns. Both the strength and longitudinal location of this low pressure center plays a strong role in determining the center and strength of the dipole in coun-

- tervailing patterns, which is qualitatively consistent with Hosking et al. (2013). Anomalous deepening of this low (represented as negative pressure changes in z500 plots, such as the pattern emerging from the basin 26-based composite, Supplementary Material) drives increased snowfall in eastern WAIS (due to enhanced onshore maritime flow) and decreased snowfall in western WAIS (due to the increased occurrence of offshore continental flow). Conversely, weakening of the ABSL promotes an opposite response (e.g., as represented by the basin 20 composite). Additionally, the longitudinal position of the ABSL plays
- an important role in determining the position and strength of countervailing dipole in snowfall variability. For example, when the ABSL is shifted to the east (as represented by the basin 22 composite) the dipole is weakened because the resulting pattern of circulation is characterized by strong southerly flow over the Weddell Sea that precludes the large increases to continental outflow over the AIS that would compensate for increased snowfall to the west. Conversely, a westerly shift in the center of variability (e.g. the basin 20/26-based composites) displays a much higher countervailing pattern in basin-integrated snow-
- fall patterns because in this case the opposite arm of the circulation center that exhibits southerly flow does extend over the Antarctic land mass.

While ABSL stands out as a dominant center of variability, it also is closely linked and often embedded within broader modes of atmospheric circulation variability that also exert broader controls on Antarctic snowfall patterns. Basin-scale compositing of several East Antarctic drainage basins, most prominently in the vicinity of Enderby Land and Dronning Maud

Land, reconstruct a notable wave-3-like difference pattern in z500. A relatively more subtle wave-three pattern also emerges in the basin-scale compositing of several AP basins and also basin 14 (George V Land) hinting at the presence of sensitivity to wave-3 variability at nodes separated by approximately 120° around Antarctica. This is consistent with the presence of a quasi-stationary, temporally variable wave-3 circulation in CESM that impacts Antarctic snowfall along southward-flowing branches of the wave train. The same pattern, at least in the Enderby Land/Dronning Maud Land region, also is associated with

coherent remote changes in z500 and moisture transport, leading to AIS-wide patterns of both muted (but significant) positive and negative inter-basin correlations, highlighting non-annularity in wave-three circumpolar circulation as an important control on countervailing inter-basin snowfall variability patterns. This finding is strongly supported by previous studies which identify the effect of wave-3 non-annularity of zonal atmospheric circulation (e.g. as quantified by the ZW3 indice, Raphael, 2004), with increasing wave-3 non-annularity closely associated with increasing meridional moisture and heat transports.

For a further set of basins, basin-scale snowfall variability appears most linked to local asymmetries in the broad zonalwave 1 atmospheric circulation. In some cases (e.g. basin 10), basins tends to receive more snowfall primarily as a result of a regional dipole of increased continental and decreased offshore z500 change. This pattern suggests the presence of a reoccurring regional elongation of zonal wave 1 asymmetry in circumpolar circulation, that is weakly reflected in more remote circulation changes. In other cases (e.g. basin 17, encompassing the South Pole), increased snowfall appears associated with a

10 circulation changes. In other cases (e.g. basin 17, encompassing the South Pole), increased snowfall appears associated with a broader pattern of higher continental and lower maritime z500 height wave 1 variability that is circumpolar in nature, and is qualitatively similar to the pattern of Southern Annular Mode (SAM) variability.

## 4 Discussion

In this study we identify the presence of significant dampening and actively countervailing tendencies in CESM-simulated

- Antarctic annually-averaged snowfall variability, the dominant term in the Antarctic surface mass balance. Our finding of countervailing Antarctic snowfall variability patterns in CESM agrees with a finding of similar patterns in RACMO2 and also in ice-core-based assessments (Thomas et al., 2017). These similarities give confidence that CESM-simulated patterns of AIS snowfall variability are physically realistic.
- Counteracting patterns of basin-integrated CESM-simulated Antarctic snowfall variability are directly linked to and driven by - variability in broad-scale atmospheric circulation patterns, which impact atmospheric moisture transport pathways. Thus, an important finding of this study is that an intrinsic characteristic of Antarctic atmospherically-controlled snowfall patterns over Antarctica is suppression of ice-sheet-integrated variability by opposing precipitation variations. Our compositing approach isolates spatial patterns of atmospheric variability that promote dampening/counteracting of the impact of variability for each ice sheet drainage basin, via simultaneous changes to other basins. These basin-specific composite patterns closely
- resemble known modes of variability that are well known regulators of Southern Hemispheric climate. These include, for example, Amundsen Sea Low variability, the Southern Annular Mode, and wave-3 non-annularity in zonal atmospheric circulation. In reality, these modes of variability are superimposed in a complex manner that likely includes inter-mode interactions (e.g. Fogt et al., 2011; Hosking et al., 2013). Yet, it is clear from the analysis presented here that each mode has a distinct signature on AIS patterns, and the combined signatures act to strongly dampen AIS integrated variability.
- In many cases, basin-specific accumulation compositing reveals similar patterns of snowfall variability for nearby basins, including similar inter-basin patterns of countervailing variability. This reflects an expected dependence of nearby basins on similar patterns of regional atmospheric variability. However, in some cases, immediately adjacent basins show no statistical correlation despite large moisture transport vector changes (for example, basins 19 and 22, Figure 4d). In these cases, lack

of correlation in snowfall variability occurs because the dominant modes of regional atmospheric variability that drive large changes in onshore moisture transport in one basin drive simultaneous changes in moisture advection that are directly crossshore over the other, resulting in a change in direction in prevailing moisture transport but insignificant net change in transport magnitude or pathway length.

- In other cases, adjacent basins exhibit not just a lack of correlation, but rather significantly anti-correlated snowfall variability. The presence of such patterns reflects the location of divides coupled to patterns of atmospheric flow anomalies. For example, increased basin 2 snowfall is correlated with decreased snowfall in multiple basins on the other side of East Antarctic ice divides, because the coherent large-scale change in atmospheric circulation that brings flow onshore over basin 2 acts simultaneously to increase continental outflow in these other locations. As a result, many Antarctic ice divides practically mark
- strong transitions in inter-basin variability phasing. This may be an additional important cause of notably decreased accumulation CV along these divides (Figure 2c), as they are influenced by synoptic systems representing counteracting modes of variability from either side of the divide over the course of a surface mass balance year.

We are confident in our qualitative conclusion that an intrinsic aspect of atmospheric circulation over Antarctica is its ability to actively dampen AIS-integrated snowfall variability. This confidence arises from corraboration of CESM results by simi-

- lar RACMO2 results, as well as indications of counteracting snowfall patterns from ice core records (Thomas et al., 2017). However, some fraction of the CESM-simulated structure of dampened and countervailing snowfall variability that CESM demonstrates may reflect model bias in both CESM-simulated mean atmospheric circulation and atmospheric circulation variability. For example, Holland et al. (2016) note a climate model bias towards too-weak modes of non-annular variability in zonal atmospheric flow (e.g. Raphael, 2004) which, if present in the the CESM LE preindustrial simulation, would spuriously
- reduce basin-scale counteracting variability. In a similar vein, the ABSL tends to exhibit notable biases in seasonal longitudinal position across a range of climate models (Hosking et al., 2013) that if present in CESM could impact spatial patterns of countervailing variability via control of ABSL location on inter-basin variability correlations. This dynamical bias, that is notably minimized in earlier simulations of CESM (Hosking et al., 2013), could potentially account for discrepancies in correlation patterns between CESM and RACMO (Figure 3) although the short RACMO time series length could also play an important

role.

We argue that countervailing tendencies in CESM-simulated Antarctic snowfall variability have a strong impact on integrated annually averaged Antarctic snowfall variability. This is quantitatively reflected in the CV of CESM-simulated AIS-integrated snowfall (0.04) being  $\sim$ 4x smaller than basin-scale averaged CV (0.17). In the counterfactual case where these dampening and countervailing patterns were entirely absent and the Antarctic continent varied synchronously with a CV of 0.17, the CESM-

- simulated AIS integrated snowfall variability would increase by  $\sim$ four-fold to a 1-sigma standard deviation of 416 Gt/yr, or approximately 1.2 mm/yr of global mean sea level (GMSL) variability. This additional variability would be comparable in magnitude to 20<sup>th</sup> century secular sea level rise rates ( $\sim$ 1.7mm/yr, Church et al. (2013)). Furthermore, it would add notably to GMSL interannual variability, which is dominated by ENSO-driven land water storage changes that generate up to  $\sim$ 5 mm of sea level anomaly (Llovel et al., 2011)). Thus, dynamically-linked dampening/counteracting Antarctic snowfall variability
- patterns play an important role in moderating GMSL changes by removing a large source of variability from the GMSL budget.

5

We also suggest that the presence of AIS snowfall variability dampening/counteracting patterns plays an important role in the detection of forced Antarctic mass change signals, because it reduces the effective noise that the signal of forced changes to Antarctic snowfall must overcome to become statistically detectable. For example, Previdi and Polvani (2016) highlight the potential of interannual Antarctic snowfall variability to mask a forced signal of snowfall increases over the 1961-2005 period. Our results suggest that the already-strong masking of forced trends by interannual variability would be even more effective.

in the case where dynamically-driven dampening/counteracting patterns in Antarctic snowfall were absent. Similarly, Wouters et al. (2013) highlight the role of Antarctic mass balance variability in obscuring assessments of ice sheet mass loss acceleration over the recent observational timeframe. Our results indicate that in the absence of dampened/countervailing Antarctic snowfall variability patterns, this already significant impediment to assessing Antarctic mass changes would be much greater.

#### 10 5 Conclusions

In this study we analyze climate model output from the Community Earth System Model Large Ensemble preindustrial control simulation to explore patterns of snowfall variability over Antarctica. Most notably, correlation analyses of CESM-simulated snowfall integrated over Antarctic ice drainage basins highlight the presence of strong patterns of dampened and even countervailing basin-integrated snowfall variability. The physical veracity of this finding is strongly supported by the presence

15 very similar behaviour by the regional RACMO model, and further confirmed by indications of countervailing precipitation in ice-core records.

Compositing of climate conditions based on the annual time series of snowfall in each basin permits investigation of the broader climate conditions associated with basin-scale integrated snowfall variability. This analysis reconstructs previously-characterized modes of Southern Hemisphere high latitude atmospheric circulation variability, confirming these modes as

- 20 dominant controllers of both basin-scale snowfall variability and inter-basin countervailing variability patterns. This control arises because of variability in atmospheric moisture transport, which is closely linked to regionally countervailing trends in Antarctic snowfall and also drives regional sea ice variability. Important modes of variability with respect to regionally countervailing patterns of Antarctic snowfall include the Amundsen/Bellingshausen Sea Low and the zonal wave three pattern in circumpolar atmospheric circulation, and variability of the Southern Annular Mode. Finally, we suggest that the presence
- of widespread dampening of snowfall variability likely has a notable impact on integrated Antarctic ice sheet mass variability, with important implications for the detection of ice sheet mass change accelerations, emergence of an anthropogenically forced signal in Antarctic integrated snowfall, and global sea level rise changes.

*Code and data availability.* All NCAR Command Language (NCL) analysis scripts (for Atmospheric Research, 2017) are available upon request to J. Fyke. Information on obtaining CESM Large Ensemble data available at http://www.cesm.ucar.edu/projects/community-projects/LENS/.