# Peer review of "Countervailing regional snowfall patterns dampen Antarctic surface mass variability"

_The Cryosphere, 2017_

## Short Comment (SC1) · 30 Jun 2017

This paper presents a nice piece of analysis of spatial patterns of coherent interannual variability of precipitation over the antarctic ice sheet and shelfs from a very long climate model preindustrial simulation. It demonstrates significant correlations and anticorrelations at inter drainage basin scales. This is of interest in its own right. However, it is not a new information that regional precipitation variations can be anticorrelated within continental scales. This occurs because the regional interannual variability of precipitation is generally associated to shifting or variable strength of major influencing synoptic systems, like the Icelandic or Azore hights affecting precipitation in Europe (north vs southern Europe). This is not new in general and not new in Antarctica. Spatially coherent (correlated or anticorrelated) patterns of variability of precipitation have previously

been exhibited by principal component analysis [e.g. Genthon et al., 2003, Interannual Antarctic tropospheric circulation and precipitation variability, Climate Dyn. 21, 298-307. DOI 10.1007/s00382-003-0329-1, and references therein] and interpreted in terms of the variability of the atmospheric circulation (500hPa geopotential, paper cited above), major driving antarctic synoptic systems and thus patterns of moisture advection towards or from Antarctica [e.g. Genthon andCosme, 2003. Intermittent signature of ENSO in west-Antarctic precipitation, Geophys. Res. Lett. 30, NO. 21, 2081, doi:10.1029/2003GL018280]. Corroboration by ice core records of such patterns have also been highlighted (e.g. Genthon et al., 2005. Interannual variability of the surface mass balance of West Antarctica from ITASE cores and ERA40 reanalyses, Climate Dyn. 24, 759-770, DOI:Âǎ10.1007/s00382-005-0019-2). I don't think that "a clear analysis of spatial heterogeneity in Antarctic snowfall variability" is really fully lacking, even though not yet necessarily in a drainage basin mode as performed here.

Further, I find it hard to buy this idea of "dampening" of continental scale precipitation variability by (anti)correlated regional variability. This sounds pretty much like, the long term global temperature trend is dampened by interannual variability which is to some extend (anti)correlated, e.g. El Nino (warm) – La Nina (cold) sequences. There is no dampening, just averaging out both uncorrelated and correlated variability.

On page 3: To confuse snow fall, precipitation and accumulation is misleading. Accumulation may be evaluated from ice cores, not precipitation. It may be a problem when tentatively comparing with ice cores. At least one of the authors is very aware that blowing snow can significantly affect the surface mass balance. Accumulation is generically used in much of the rest of the paper even including in section 3.3 where ice cores are discussed, although P-E is finally used in section 4.4.. I rather suggest to stick to precipitation when it is precipitation, surface mass balance when it is surface mass balance, P-E when it is precip – evap.

Section 3.5 on 'links to broader patterns of atmospheric variability" falls a bit short. A contribution of the wave-3 pattern and SAM are mentioned but not really demonstrated,

e.g. by calculating the time correlation between indices of these modes in the model and precipitation anomalies. The Amundsen – Belinghausen low is definitely a major center of natural variability in the region, directly affecting moisture advection to and from the Antarctica, but the fact that this is related to both the SAM and the ENSO, broader patterns of variability, could also probably be reported (Genthon et al., 2003, see above).
* * *

---

## Referee Comment (RC1) · Anonymous Referee #1 · 28 Jul 2017

General comments:

This paper assesses natural variability in Antarctic snowfall patterns using output from a long CESM pre-industrial control simulation. It is determined that different Antarctic regions – delineated using ice drainage basins – exhibit out-of-phase snowfall anomalies that lead to a dampening of the overall ice sheet-integrated snowfall variability. This has implications for the variability of global mean sea level, and the detection of anthropogenically-forced changes in Antarctic mass balance.

This is a nice study that represents a valuable addition to the literature on Antarctic mass balance variability and change. While the main result – out-of-phase snowfall anomalies in different parts of Antarctica – is not terribly surprising or novel, it is worth documenting in a systematic way as the authors have done, using a basin-scale anal-

ysis and a long (1800-year) model control simulation (these latter aspects of the study are, I believe, novel). The paper is clear and well written, and the conclusions drawn are supported by the analyses presented. While I believe that the manuscript can basically be published in its present form, I include below several specific comments and technical corrections that the authors may wish to consider.

Specific comments:

1) p. 5, lines 25-33: It would be helpful to give the spatial correlation between the CESM and RACMO2 snowfall variability patterns (i.e., the correlation between the matrices in Figs. 3a and 3b).

2) p. 7, lines 28-33: This might be a good place to remind the reader that on annual timescales P-E is essentially equivalent to moisture flux convergence, since changes in atmospheric moisture storage can be neglected.

3) p. 8, lines 20-26: The last two sentences of this paragraph are confusing to me. First, it is stated that the basin 22 composite depicts an eastward shift of the ABSL. This looks like a westward shift to me. Second, for this same composite, it is stated that "strong southerly flow over the Weddell Sea...precludes the large increases to continental outflow over the AIS". In contrast, I would think of southerly flow over the Weddell Sea as being associated with continental outflow, not precluding it. Finally, it is stated that the basin 20/26 composites depict a westward shift of the ABSL. I don't see this; to me, these composites suggest a weakening/strengthening of the ABSL (rather than a shift in position), and, in fact, this is how they are described in the same paragraph above.

4) p. 10, lines 18-25: It seems to me that it would be easy enough to check whether these biases are actually present in the CESM LE preindustrial simulation, rather than just speculating.

Technical corrections:
[Figure]

1) p. 1, line 12: "AIS" not yet defined.

2) p. 3, line 17: The word "also" appears twice.

3) p. 3, lines 30-31: I suggest removing "climatological" from the phrase "standard deviation of the climatological annual-averaged accumulation".

4) p. 4, line 11: "WAIS" not yet defined.

5) p. 19, Fig. 3 caption: "Significance shaded at the 95% level." – Would be clearer (and consistent with subsequent figures) to say that stippling indicates significance at the 95% level.

6) p. 5, line 15: Should be "for a set".

7) p. 5, line 16: "Analogous" is misspelled.

8) p. 6, lines 24-25: Should be "Figure 4".

9) p. 7, line 6: "EAIS" not yet defined.

10) p. 22, Fig. 6 caption: "arrows as in Figure 4" – I don't see arrows in Figure 6.

11) p. 7, line 28: "P-E" not yet defined.

12) p. 9, line 4: Should be "ZW3 index".

13) p. 9, line 7: Should be "basins tend to receive".

14) p. 10, line 5: Should remove the word "just" from this sentence.

15) p. 10, line 14: The word "corroboration" is misspelled.

16) p. 10, line 19: The word "the" appears twice.

17) p. 10, line 31: Would be better to say "This greater variability".

18) p. 11, lines 14-15: I suggest "supported by the presence of very similar behaviour in the regional RACMO model".

19) p. 11, line 22: I'd remove the word "drives".

20) p. 11, line 22: I'd suggest instead "Important sources of variability".

---

## Author Comment (AC1) · 1 Aug 2017

To Dr. Genthon,

First, many thanks for your valuable thoughts on our manuscript. Particularly, we appreciate your identification of previous papers that are directly relevant to our study, and so should have been referenced and discussed. We will do this in a revised version of this manuscript. Below, we attempted to extract your main points (quoted) for further discussion. After reading our replies, please do not hesitate to continue the discussion as you see fit.

"[Correlated/anti-correlated regional precipitation patterns] at continental scales is not a new information. . . this occurs because the regional interannual variability of pre-

cipitation is generally associated to shifting or variable strength of major influencing synoptic systems. This is not new ... in Antarctica."

We agree that the general concept of (anti)-correlated regional precipitation patterns at continental scales is not new, even for Antarctica. We will certainly emphasize this further and reference your indicated studies, which are important for our study and which we apologize for missing. Regardless, we believe our contribution (i.e., basin-scale disaggregation of precipitation variability from the 1800-year record of unforced Antarctic climate in the coupled climate model) is still significant, largely because it provides a clean evaluation of continent-wide basin-specific natural Antarctic spatial variability that is by definition uncontaminated by anthropogenically forced signals (unlike reanalyses and/or ice cores, which are nonetheless valuable in their own right).

"Spatially coherent (correlated and anticorrelated) patterns of variability of precipitation have previously been exhibited by principal component analysis and interpreted in terms of the variability of the atmospheric circulation, major driving Antarctic synoptic systems and thus patterns of moisture advection towards of from Antarctica. Corroboration by ice cores has also been highlighted."

Thanks - we should definitely note these analyses in our manuscript to identify prior findings linking snowfall variability to atmospheric circulation. Building on these results, we suggest that our basin-scale compositing approach to identify important patterns of atmospheric circulation is still an important and novel advance, because it uniquely extracts from a millennial-scale climate model simulation the characteristic patterns of atmospheric circulation/moisture advection that are specific to individual ice sheet drainage basins. With this approach, regional differentiations in important atmospheric circulation modes emerge, in a manner not possible with continental-scale PCA analysis that is dominated by a few major modes (e.g. the ABSL). Additionally, our model-based, basin-scale approach is able to explicitly identify a lack of sea ice control on regional Antarctic snowfall variability, which has been posited as important in other recent studies.

[Figure]

"I don't think that a 'clear analysis of spatial heterogeneity in Antarctic snowfall variability' is really fully lacking."

Particularly after being pointed to your earlier work, we agree, and will temper our statements accordingly in the revised manuscript.

"I find it hard to buy this idea of "dampening" of continental scale precipitation variability by (anti)correlated regional variability."

We are not sure the analogy of ENSO temporal variability dampening a long-term global temperature trend (i.e. a forced response) applies to the argument we make regarding dampening of AIS spatial variability by regionally opposing variability signals. Or perhaps we just misunderstand your comment, in which case, please feel free to let us know how we're getting it wrong! Specifically, in the ENSO case, it is true that La Nina and El Nino are in a sense anticorrelated. However, this anticorrelation occurs with a multi-year time lag so that, for example, in terms of overall variability, an El Nino doesn't 'cancel' a La Nina. Conversely, the presence of opposing Antarctic snowfall variability at different locations with zero time lag, indicates that these regional patterns are cancelling, in their impact on AIS-integrated snowfall variability (and thus sea level variability). Furthermore, statistical significance of these opposing variability signals indicates that the dampening is not simply an effect of averaging random signals, but rather (at least in part) due to active cancellation of correlated signals associated with variability in large-scale atmospheric moisture transport.

"To confuse snow fall, precipitation and accumulation is misleading."

We agree, and will rephrase our manuscript to be more accurate.

"Links to broader patterns of atmospheric variability falls a bit short."

We welcome this comment, and pending reviewer comments we will consider how to improve this aspect of our study in the revised version.

---

## Short Comment (SC2) · 9 Aug 2017

Thank you for your replies.

Concerning this issue: "We are not sure the analogy of ENSO temporal variability dampening a long-term global temperature trend (i.e. a forced response) applies to the argument we make regarding dampening of AIS spatial variability by regionally opposing variability signals."

Sorry, I'd better stick to spatial variability issues to better explain my argument, so lets use the NAO instead. Variations of the NAO associate with opposite (anti correlated) precipitation anomalies in southern / northern Europe. Can we conclude that the impact of the NAO dampens the averaged variability in the region? It spatially modulates,

but the overall variability is what it is. The title as it stands and part of the text convey the idea that the overall precip variability over Antarctica would be larger if similar processes that induce spatially anticorrelated modulation were not at work. It is probably just a semantic issue – and may be even just poor appreciation of English subtleties on my side!

---

## Referee Comment (RC2) · Anonymous Referee #2 · 7 Sep 2017

This paper provides new insights into regional variability in Antarctic snowfall from the analysis of an 1800 year preindustrial control simulation from CESM. The paper is well-written, with datasets and methodology clearly described. Results are generally clearly presented, compared to previous findings, with carefully supported conclusions. The paper can be accepted for publication following consideration of the minor points raised below:

1. It is not clear in the text and figure captions (e.g. Figure 3) what periods of data are being used e.g. in Fig. 3 are you comparing average spatial correlations from 1800 years versus 1979-2015 from RACMO2?

2. Related to point 1 above, is there any evidence of multi-decadal variability in the spatial correlation patterns in the CESM control simulation? One would expect to see

some evidence of pattern shifts related to periodic shifts in circulation that are a characteristic of the climate system. If there is, are there particular periods that favour increased/decreased Antarctic-averaged snowfall?

3. The issue of multi-decadal variability also comes into the statement made at the top of page 6 about "sufficient record length"... if there is significant multi-decadal variability in spatial correlation structure, long-term averaging may mask an important source of temporal variabiliy in ice-sheet-integrated accumulation.

4. The sentence on page 9, line 20 "Thus, an important .... by opposing preciptiation variations" is difficult to follow. The first "Antarctic" is redundant, and it looks like something is missing after "ice-sheet-integrated variability" (in accumulation?).

5. It is nitpicking, but I found there was a tad too much repetition of your central theme of contervailing trends throughout the paper.

---

## Author Response (AR1)

**The impact of regional precipitation patterns on Antarctic surface mass variability Response to Reviewers**

Jeremy Fyke Jan Lenaerts Hailong Wang

September 2017

We thank all reviewers for their valuable comments. In addition we acknowledge Christophe Genthon in the manuscript for his short comments via The Cryosphere Discussions framework. Below we provide replies to all Reviewer comments (identified in *blue*). For the Editor's benefit we have also additionally included replies to Dr. Genthon's comments, in light of their applicability. In all cases we included indications of how the text has been revised, which can also be found via mark-ups in the attached manuscript copy.

**1 Replies to Reviewer 1 comments**

**1.1** Specific comments**

p. 5, lines 25-33: It would be helpful to give the spatial correlation between the CESM and RACMO2 snowfall variability patterns (i.e., the correlation between the matrices in Figs. 3a and 3b).

We find a linear regression coefficient of 0.82 between the strength of CESM and RACMO inter-basin correlations, highlighting the similar spatial pattern of snowfall relationships between the two models. This is now noted in the text.

p. 7, lines 28-33: This might be a good place to remind the reader that on annual timescales P-E is essentially equivalent to moisture flux convergence, since changes in atmospheric moisture storage can be neglected.

We have now added text to this effect.

p. 8, lines 20-26: The last two sentences of this paragraph are confusing to me. First, it is stated that the basin 22 composite depicts an eastward shift of the ABSL. This looks like a westward shift to me. Second, for this same composite, it is stated that "strong southerly flow over the Weddell Sea…precludes the large increases to continental outflow over the AIS". In contrast, I would think of southerly flow over the Weddell Sea as being associated with continental outflow, not precluding it. Finally, it is stated that the basin 20/26 composites depict a westward shift of the ABSL. I don't see this; to me, these composites suggest a weakening/strengthening of the ABSL (rather than a shift in position), and, in fact, this is how they are described in the same paragraph above.

We agree that this dense text is confusing. Since it is basically secondary to our study, we simply removed these both sentences.

p. 10, lines 18-25: It seems to me that it would be easy enough to check whether these biases are actually present in the CESM LE preindustrial simulation, rather than just speculating.

We have clarified this text substantially. We now note that Holland et al. (2016) indicate a weakened CESM non-annular variability, including ABSL variability. This bias is consistent with our finding of weaker CESM correlations relative to RACMO. We now note this in the text. Regarding longitudinal position, we now note that Hoskins et al. (2013) and Holland et al. (2016) both indicate a reasonably simulated ABSL longitudinal position in CESM. As a result, we now argue in the revised text that this potential bias is not a significant impactor of our results.

**1.2** Technical corrections**

p. 1, line 12: "AIS" not yet defined. Fixed.

p. 3, line 17: The word "also" appears twice. Fixed.

p. 3, lines 30-31: I suggest removing "climatological" from the phrase "standard deviation of the climatological annual-averaged accumulation". Fixed.

p. 4, line 11: "WAIS" not yet defined. Fixed.

p. 19, Fig. 3 caption: "Significance shaded at the 95% level." – Would be clearer (and consistent with subsequent figures) to say that stippling indicates significance at the 95% level. Fixed.

p. 5, line 15: Should be "for a set". Fixed.

p. 5, line 16: "Analogous" is misspelled. Fixed.

p. 6, lines 24-25: Should be "Figure 4". Fixed.

p. 7, line 6: "EAIS" not yet defined. Fixed.

p. 22, Fig. 6 caption: "arrows as in Figure 4" - I don't see arrows in Figure
6. 11) Fixed.

p. 7, line 28: "P-E" not yet defined. Fixed.

p. 9, line 4: Should be "ZW3 index". Fixed.

p. 9, line 7: Should be "basins tend to receive". Fixed.

p. 10, line 5: Should remove the word "just" from this sentence. Fixed.

p. 10, line 14: The word "corroboration" is misspelled. Fixed.

p. 10, line 19: The word "the" appears twice. Fixed.

p. 10, line 31: Would be better to say "This greater variability". Fixed.

p. 11, lines 14-15: I suggest "supported by the presence of very similar behaviour in the regional RACMO model". Fixed.

p. 11, line 22: I'd remove the word "drives". Fixed.

p. 11, line 22: I'd suggest instead "Important sources of variability". Fixed.

**2 Replies to Reviewer 2 comments**

It is not clear in the text and figure captions (e.g. Figure 3) what periods of data are being used e.g. in Fig. 3 are you comparing average spatial correlations from 1800 years versus 1979-2015 from RACMO2? We have now clarified to indicate that we are contrasting 1800 years of CESM control simulation with years 1979-2015 from RACMO.

Related to point 1 above, is there any evidence of multi-decadal variability in the spatial correlation patterns in the CESM control simulation? One would expect to see some evidence of pattern shifts related to periodic shifts in circulation that are a characteristic of the climate system. If there is, are there particular periods that favour increased/decreased Antarctic-averaged snowfall?

The suggestion to explore multi-decadal variability of inter-basin correlations is an excellent one. To this end we performed evolving basin correlations within a 31-year moving window, for the entire 1800-year CESM simulation. Results of this analysis are presented in a new Figure 7 and several new paragraphs, which describe evolution of positive/negative/weak correlations between basin 19 and other basins. These particular comparisons were objectively chosen based on analysis of the correlation time series to identify the strongest positive/negative mean correlations, and the correlation time series whose range spanned zero and demonstrated the largest range of all such inter-basin correlations. In each case, basin 19 emerged as a good example to present.

Using this example we demonstrate the presence of large, low frequency variability in correlation strengths, particularly in negative and weak inter-basin correlations. For example, we find that the long-term weakly negative (but significant) basin 4/19 correlation masks coherent periods where the correlation is strongly negative, and other periods where it is remarkably notably positive. Thus, while the long-term correlations that form the basis for this study are still certainly valid, we now elaborate on the potential for large variations in correlation over long timescales to deviate from the long-term average correlation.

The issue of multi-decadal variability also comes into the statement made at the top of page 6 about "sufficient record length"... if there is significant multi-decadal variability in spatial correlation structure, long-term averaging may mask an important source of temporal variability in ice-sheet-integrated accumulation.

As above. We also note in the revised manuscript how this natural lowfrequency variability in CESM-simulated inter-basin correlations is perhaps closely reflective of observed low-frequency changes in ice cores and likely impacts historical estimates of snowfall variability. This is a very interesting connection to make.

The sentence on page 9, line 20 "Thus, an important .... by opposing precipitation variations" is difficult to follow. The first "Antarctic" is redundant, and it looks like something is missing after "ice-sheet-integrated variability" (in accumulation?).

Fixed.

It is nitpicking, but I found there was a tad too much repetition of your

**central theme of contervailing trends throughout the paper.**

We are in agreement, and have removed a number of redundant references to 'countervailing' and similar throughout the text. More importantly, we have altered the title to reflect this comment - the new title we feel being more objective and less focussed on only one aspect (countervailing precipitation patterns) of the study.

**3** Additional replies to Dr. Genthon comments**

Correlated/anti-correlated regional precipitation patterns at continental scales is not a new information. . . this occurs because the regional interannual variability of precipitation is generally associated to shifting or variable strength of major influencing synoptic systems. This is not new . . . in Antarctica."

We agree that the general concept of (anti)-correlated regional precipitation patterns at continental scales is not new for Antarctica. We now emphasize this further, particularly by referencing Dr. Genthon's important studies. Regardless, we also highlight more prominently that our contribution (i.e., basin-scale disaggregation of precipitation variability from the 1800-year record of unforced Antarctic climate in the coupled climate model) is still significant, largely because it provides a clean evaluation of basin-specific natural Antarctic spatial variability that is by definition uncontaminated by anthropogenically forced signals (unlike reanalyses and/or ice cores, which are nonetheless valuable in their own right) and is not impacted by the cancellation of regional modes of variability, nor by the 'swamping' of regionally important but subtle East Antarctic variability signals by more prominent West Antarctic variability modes.

Spatially coherent (correlated and anticorrelated) patterns of variability of precipitation have previously been exhibited by principal component analysis and interpreted in terms of the variability of the atmospheric circulation, major driving Antarctic synoptic systems and thus patterns of moisture advection towards of from Antarctica. Corraboration by ice cores has also been highlighted.

We now highlight these important analyses in our manuscript to identify prior findings linking snowfall variability to atmospheric circulation. Building on these results, our basin-scale compositing approach to identify important patterns of atmospheric circulation is still an important and novel advance, because it extracts from a millennial-scale climate model simulation the characteristic patterns of atmospheric circulation/moisture advection that are specific to individual ice sheet drainage basins. This information is not available from previous EOF-based studies, even those carried out over limited Antarctic domains. Additionally, our model-based, basin-scale approach is able to explicitly identify a lack of sea ice control on been posited as important in other recent studies.

I don't think that a 'clear analysis of spatial heterogeneity in Antarctic snowfall variability' is really fully lacking.

Particularly after being pointed to Dr. Genthon's earlier work, we agree that it is not 'fully lacking', and have tempered our statements accordingly in the

**revised manuscript.**

**I find it hard to buy this idea of "dampening" of continental scale precipitation variability by (anti)correlated regional variability.**

After considering this comment (and related discussions, available on the TCD website) we continue to suggest that anti-correlated regional variability does indeed act to dampen continental-scale precipitation variability. However, we have highlighted more strongly that this dampening is relative to a *counterfactual* case where all of Antarctic snowfall varies to the same variability forcing.

**To confuse snow fall, precipitation and accumulation is misleading.**

We have carefully rephrased the manuscript to be more accurate (we now refer to all model results as 'precipitation' and refer to 'accumulation' when discussing observations of net surface mass gain). We also included a brief footnote to clarify all terms.

**Links to broader patterns of atmospheric variability falls a bit short.**

We have attempted to strengthen this aspect of the study by better referencing previous connections between Antarctic circulation and broader patters of variability. However, given our focus in this manuscript is primarily to quantify inter-basin snowfall variability relationships, we feel that a comprehensive examination of large-scale climate controls on Antarctic snowfall - a very complex subject - is beyond the scope of the initial evaluation we have already provided.

**Countervailing regional snowfall patterns dampen Basin-scale heterogeneity in Antarctic precipitation and its impact on surface** mass variability**

Jeremy Fyke1, Jan T. M. Lenaerts2,3, and Hailong Wang4

1Los Alamos National Laboratory

2Department of Atmospheric and Oceanic Sciences, University of Colorado, Boulder, USA
 3Institute for Marine and Atmospheric Research, Utrecht University, Utrecht, Netherlands
 4Pacific Northwest National Laboratory, Richland, USA

Correspondence to: Jeremy Fyke (fyke@lanl.gov)

Abstract. Snowfall over AntarcticaPrecipitation in the form of snow, the dominant term of the Antarctic ice sheet surface mass balance, displays large regional heterogeneity in temporal variabilitypatterns. This heterogeneity has the potential to dampen variability in integrated Antarctic surface mass trends by counteracting increases in snowfall in one location with decreases in another (and vice versa). To examine the presence of countervailing regional snowfall patterns, here spatial and temporal

- 5 variability. Here we present an analysis of spatial patterns of regional Antarctic snowfall variability, their broader climate drivers precipitation variability and their impact on integrated Antarctic snowfall surface mass balance variability simulated as part of a preindustrial 1800 year equilibrated global, fully coupled Community Earth System Model simulation. Correlation and composite analyses based on this output allow for a statistically robust exploration of Antarctic snowfall precipitation variability. We uncover statistically significant countervailing snowfall-identify statistically significant relationships between
- 10 precipitation patterns across Antarctica that are corroborated by climate reanalyses, regional modelling and ice core records. These countervailing patterns are driven by variability in large-scale atmospheric moisture transportand cause large spatial heterogeneity in temporal variability, with, which itself is characterized by decadal to centennial scale oscillations around the long term mean. We suggest that this heterogeneity in Antarctic precipitation variability has a dampening effect on overall Antarctic snowfall variabilitymagnitude. surface mass balance variability, This dampening has with implications for regula-
- 15 tion of Antarctic-sourced sea level variability, detection of an emergent anthropogenic signal in Antarctic mass trends and identification of AIS-Antarctic mass loss accelerations.

**1 Introduction**

Precipitation of snow over Antarctica is the means by which the Antarctic Ice Sheet (AIS), currently the largest contiguous ice complex on the planet, accumulates gains mass. Over millenniatime, this accumulation has resulted in the sequestration of 57m

20 of eustatic sea level equivalent as AIS ice (Fretwell et al., 2013). As a result of the large area of this capacitor of freshwater, natural variability in AIS-integrated snowfall precipitation has a leading impact on AIS mass trends, which in turn influence global mean sea level (GMSL) variability.

AIS-integrated accumulation variability is the net precipitation variability combines the effect of regional accumulation precipitation variability signals. Previous studies have leveraged Gravity Recovery and Climate Experiment (GRACE) output, satellite altimetry and/or regional modeling to highlight strongly heterogeneous spatial patterns of temporal variability (Mémin et al., 2015; Martín-Español et al., 2016; Shepherd et al., 2012; Boening et al., 2012)in surface mass balance (SMB)

5 (e.g. Mémin et al., 2015: Martín-Español et al., 2016: Shepherd et al., 2012; Boening et al., 2012). Complementing these remotelysensed observations, longer pointwise time series of accumulation variability SMB from ice cores (.e.g. Frezzotti et al., 2013; Thomas et al., 2017, and references therein) also suggest the presence of heterogeneous regional patterns of snowfall precipitation variability. Historical climate reanalysis (Dee et al., 2011) as well as regional models driven by this reanalysis (e.g. Van Wessem et al., 2014) further confirm the presence of high spatial heterogeneity in recent historical Antarctic accumulation

10 variability trends (Wang et al., 2016) precipitation variability signals (Genthon et al., 2003; Genthon and Cosme, 2003; Wang et al., 2016)

These analyses provide critical insight into historical drivers of ice sheet mass balance variability and hint at a potentially strong role for heterogenous patterns of snowfall variability in dampening heterogeneous patterns of precipitation variability in regulating AIS-integrated snowfall and associated ice sheet mass variability. They are also important in understanding future

- 15 change, during which regional variability will be superimposed on by a strong secular trend towards greater accumulation precipitation (Palerme et al., 2016; Frieler et al., 2015; Previdi and Polvani, 2016; Lenaerts et al., 2016) and increased lowelevation surface melting (Fyke et al., 2010; Trusel et al., 2015). However, they are characteristically hampered by either short time series length, sparse point data, observational uncertainty, a mixture of forced and natural variability signals, or a combination of all of these. As a result, a clear analysis of spatial heterogeneity in Antarctic snowfall variability – and the
- 20 impact of this heterogeneity on AIS-integrated snowfall variability and associated mass change is currently lacking. Analysis of global climate model output provides a potential pathway around these difficulties. In particular, pre-industrial preindustrial climate model control simulations - long quasi-equilibrium simulations in which no time-varying external forcing is applied (e.g. Eyring et al., 2016) - provide spatially complete, temporally extensive representations of natural climate variability that are uncontaminated by external forcing changes. Motivated by the potential for climate model output to provide
robust dynamical insight into drivers of spatially heterogeneous Antarctic accumulation precipitation variability, here we use the Community Earth System Model (Hurrell et al., 2013) to explore spatial patterns in Antarctic snowfall precipitation variability.

ability. The study is presented as follows: in Section 2 we describe the model, simulation and analysis methods we adopt to explore Antarctic accumulation precipitation variability. In Section 3 we detail the findings of our analyses, which are further interpreted in Sections 4 and 5.

**2 Methods**

We explore AIS basin-scale accumulation precipitation1 variability and its link to broader climate patterns using output from the Community Earth System Model (CESM, version 1). CESM is a comprehensive coupled Earth system model that is used for a diverse range of applications, including recent explorations of Southern Hemisphere elimate (Holland et al., 2016; Lenaerts et al., 2016)

- 5 and Antarctic climate (e.g. Holland et al., 2016; Lenaerts et al., 2016). The particular CESM simulation we focus our analysis on is a ~1 degree resolution preindustrial control simulation of the CESM Large Ensemble experiment (CESM LE hereafter) (Kay et al., 2015). Output from years 400-2200 of this fully coupled simulation provides a robustly equilibrated (and thus statistically stationary) representation of global climate system dynamics that is uncontaminated with anthropogenic or other external climate forcings, and of sufficient length (1800 years) to support robust variability analysis. These benefits
- 10 provide an important advantage of this approach, over analogous studies based on reanalysis data This approach to assessing Antarctic variability provides important insights that are difficult to obtain using reanalysis products from the relatively shortand/non-stationary recent historical period. Our methodological approach is similar to other investigations of unforced Southern Hemisphere climate variability , that also that use output from climate model preindustrial control simulations (e.g. Santoso et al., 2006; Holland et al., 2005; Previdi and Polvani, 2016)control simulations forced with either constant preindustrial
- 15 or constant present day conditions (e.g. Genthon and Cosme, 2003; Santoso et al., 2006; Holland et al., 2005; Previdi and Polvani, 2016)

In our analysis, we We represent regional Antarctic accumulation precipitation heterogeneity by disaggregating the AIS into individual AIS drainage basins (Zwally et al., 2012) (Figure 1). Ice shelves are included in this basin disaggregation. This approach is adopted because each basin represents a distinct dynamical integrator of regional climate signals, making a basin-scale

20 analysis a practical and glaciologically tractable method for linking climate-driven snowfall-precipitation variability to regional Antarctic glaciological trends. ice dynamical changes. Previous Antarctic principal component analysis-based precipitation studies have identified major benefits associated with use of limited Antarctic (Genthon et al., 2003) and West Antarctic ice sheet (WAIS) (Genthon and Cosme, 2003) domains. Specifically, these domains allow for isolation of important sector-relevant variability patterns that can be obscured within of larger-scale hemispheric or global analyses. By further extending the 'local

25 analysis' methodology to individual ice sheet basins, we provide even greater control on identification of regionally critical variability drivers, which in larger-scale approaches can overlap and also be obscured by more dominant variability spatial modes (Genthon et al., 2003).

At the basin scale, we focus primarily on drivers of variability in annually-averaged, spatially integrated precipitation rates (units of Gt/yr) because of the importance of this measure for mass balance and dynamics, combined with the relative in-

30 sensitivity of long-term AIS ice sheet dynamics to higher-frequency variability (e.g. related to the seasonal cycle and/or individual storm events, which nonetheless are important in summing to the net annual signal). We apply several analysis tools to quantify simulated basin-scale snowfall variability, compare inter-basin variability relationships, and link basin-scale

<sup>1In the following, we consider 'precipitation' to be practically synonymous with 'snowfall', since rain accounts for approximately 0.1% of total precipitation over Antarctica in the simulation described here. We define 'accumulation' as SMB, which for much of Antarctica equals precipitation minus sublimation.

variability to broader patterns of climate variability. We first perform correlation analysis between basin-specific annuallyaveraged accumulation precipitation time series to identify inter-basin accumulation-variability relationships. In addition, for each basin, we construct use composite climatologies based on CESM-simulated annually-averaged accumulation basin-scale annually averaged precipitation to construct paired monthly-resolved characteristic climate states associated with high/low

- 5 basin accumulation precipitation years. This allows examination of the relationships between basin accumulation timeseries basin-resolved precipitation time series and broader modes of climate variability. To verify that the patterns of variability identified in CESM model output were are robust, we repeated subsets of our analysis on output from a 27 kilometer resolution RACMO2.3 regional atmosphere/ice sheet surface model simulation spanning the 1979-2015 period. RACMO2.3 has been extensively applied and validated over Antarctica, thus providing an important benchmark of CESM performance (Lenaerts
- 10 et al., 2016). Finally, we also also assess the similarity of compare the CESM results against compilations estimates of accumulation variability derived from ice core records, and identify the impact of low frequency variability on Antarctic precipitation patterns by performing time-dependent basin-scale correlation analysis.

**3 Results**

**3.1 Simulated preindustrial Antarctic accumulation precipitation variability**

[revised manuscript text omitted]
 precipitation years co-occurring with positive local on-shore moisture transport anomalies and low accumulation precipitation years co-occurring with weaker on-shore transport anomalies (Figure 5 4 and Supplemental Material). Within each basin, the difference in absolute moisture flux tends to weaken inland, reflecting the dominance of coastal orographically-driven precipitation in Antarctica (Favier et al., 2013). Given that basin-integrated accumulation precipitation mostly reflects coastal precipitation, the compositing technique prefer-
- 30 entially clusters years with greater coastal precipitation, leading to a tendency towards composite differences that are coastally maximized maximized towards the coast. Moisture flux anomalies are often largest in the local vicinity of the basin for which the compositing was based, identifying local circulation variability as an important final link between remote evaporative source regions and final precipitation location.

The tight relationship between Antarctic basin-scale accumulation-precipitation and proximal moisture transport variability confirms a leading role for moisture transport in determining local basin snowfall variability. However, at a broader precipitation variability (Genthon et al., 2003). At a larger scale, the composite climatologies also identify spatially coherent large scale confirm spatially coherent moisture transport changes as the causal factor behind interbasin countervailing accumulation

- 5 inter-basin countervailing precipitation variability. Specifically, for all basins that demonstrate simultaneous positive/negative accumulation-precipitation correlations with the basin for which compositing procedure is based on, the compositing also identifies corresponding onshore/offshore moisture transport anomalies. As an arbitrary demonstrative example, each basin with snowfall-precipitation variability that is positively correlated with basin 7 (Figure 4b) also demonstrates increased on-shore moisture transport, with the strength of the correlation tending to increase with increased onshore moisture transport
- 10 changes. Such basins include several proximal basins, as well as remote basins on the AP and in Victoria land. The converse relationship holds for anti-correlated basins that tend to occur as a swath of basins across the interior of the EAIS. In general, this relationship between basin accumulation correlations and local and remote moisture transport changes holds across all basin-specific composites, strongly implicating large-seale modes of polar atmospheric moisture advection as the active driver of countervailing basin-seale accumulation variability. East Antarctic ice sheet.
- 15 Moisture flux anomalies associated with basin-specific accumulation precipitation variability, as well as broader patterns of interbasin dampening/countervailing accumulation-precipitation variability, are tightly linked to composited differences in both 500 hPa geopotential height (z500, contours in Figure 5) and sea level pressure (SLP, not shown). Stemming from the similarity between SLP and z500 composite differences (r=0.91) we limit\_focus our attention to the relationship of moisture flux changes to z500 variability, which provides a consistent metric of atmospheric circulation both at sea level and over
- 20 the elevated ice sheet topography (Genthon et al., 2003). Basin-based accumulation precipitation compositing reveals spatially extensive and significant z500 differences with a transition from positive to negative anomalies indicating large changes in horizontal pressure gradients gradient anomalies typically lying in close proximity to the basin from which compositing was based (see, for example, Figure 5a, b and d). Moisture flux anomalies, which are characteristically maximal close to the composited basin, are consistently ~90 degrees leftwards of the maximum z500 gradient. This points to regional geostrophic
- 25 atmospheric variability flow as a main determinant of the moisture transport variability that, which in turn drives inter-annual basin-scale accumulation precipitation variability.

Despite However, despite the apparently robust control of atmospheric circulation variability on Antarctic basin-scale snowfall precipitation variability, another potential regulator of atmospheric moisture transport to Antarctica must also be considered, namely, regional sea ice concentration (SIC)-regulated local evaporative source region variability (Tsukernik

- 30 and Lynch, 2013; Thomas and Bracegirdle, 2015). Compositing reveals that basin-scale accumulation variability is typically precipitation variability is closely associated with nearby offshore SIC variability: basin-specific high-accumulation high-precipitation years tend to be characterized by statistically significant local decreases in upwind\_offshore\_SIC, while low-accumulation low-precipitation years are characterized by upwind-offshore\_SIC increases (Figure 6). The broad co-occurrence of low/high upwind SIC with downstream high/low basin-scale accumulation\_precipitation\_suggests a role for sea ice-regulated local
- 35 evaporation source variability in determining basin-scale accumulation precipitation variability. However, we argue against

this causal relationship, because strong increases in annual precipitation minus evaporation (P-E, approximately equivalent to moisture flux convergence on annual timescales) that are proximal to the basin upon which compositing was performed are *not* not associated with opposing regions of negative P-E change over proximal nearby sea ice loss regions (Figure 6). Instead, P-E decreases over these regions are typically weak and in some cases even positive (e.g. Figure 6a/b). This

- 5 contradicts, contradicting the pattern that would be expected if SIC-regulated evaporative increases were a dominant factor in locally increased Antarctic basin-scale snowfallprecipitation: namely, a strong proximal offshore P-E decrease (reflecting increased evaporation that is not countered by opposing over-ocean precipitation) closely associated with the pattern of negative SIC change next to basins experiencing increased snowfallprecipitation. The argument for a weak dependence of AIS snowfall Antarctic precipitation on local SIC variability state also holds for the case of basins experiencing countervailing
- 10 snowfall\_decreased precipitation but with reversed tendencies of change. Thus, we conclude that regional sea ice variability is not the primary factor driving basin-scale Antarctic snowfall\_precipitation variability. Rather, both sea ice and Antarctic snowfall\_precipitation variability are both commonly regulated by atmospheric circulation variability, which leads to the appearance of similar-associated (but not causal) spatial variability patterns. This conclusion contradicts speculations by Tsukernik and Lynch (2013); Thomas and Bracegirdle (2015); Lenaerts et al. (2016) Tsukernik and Lynch (2013); Thomas and Bracegirdle (2015); Lenaerts et al. (2016) Tsukernik and Lynch (2013); Thomas and Bracegirdle (2015); Lenaerts et al. (2016) Tsukernik and Lynch (2013); Thomas and Bracegirdle (2015); Lenaerts et al. (2016) Tsukernik and Lynch (2013); Thomas and Bracegirdle (2015); Lenaerts et al. (2016) Tsukernik and Lynch (2013); Thomas and Bracegirdle (2015); Lenaerts et al. (2016) Tsukernik and Lynch (2013); Thomas and Bracegirdle (2015); Lenaerts et al. (2016) Tsukernik and Lynch (2013); Thomas and Bracegirdle (2015); Lenaerts et al. (2016) Tsukernik and Lynch (2013); Thomas and Bracegirdle (2015); Lenaerts et al. (2016) Tsukernik and Lynch (2013); Thomas and Bracegirdle (2015); Lenaerts et al. (2016) Tsukernik and Lynch (2013); Thomas and Bracegirdle (2015); Lenaerts et al. (2016) Tsukernik and Lynch (2013); Thomas and Bracegirdle (2015); Lenaerts et al. (2016) Tsukernik and Lynch (2013); Thomas and Bracegirdle (2015); Lenaerts et al. (2016) Tsukernik and Lynch (2017); Thomas and Bracegirdle (2017); Thomas and Bra
- 15 and Lenaerts et al. (2016) but supports the findings of Singh et al. (2016) and Genthon et al. (2003) and Singh et al. (2016) and also recent studies exploring sea ice controls on Greenland Ice Sheet climate (Noël et al., 2014; Stroeve et al., 2017).

**3.5 Links to broader patterns of atmospheric variability**

Establishing broad-scale atmospheric circulation variability as the main determinant of both basin-scale snowfall variability, and broader patterns of precipitation variability and broader inter-basin variability dampening and cancellation, subsequently

20 relationships allows for a broader assessment of the role of larger-scale atmospheric circulation variability in controlling these features. Notably, the pattern of atmospheric circulation differences extracted via compositing of basin-scale snowfall clearly reconstructs with remarkable fidelity precipitation clearly reconstructs previously-characterized broad-scale patterns of variability, highlighting their important impact.

The CESM-simulated Amundsen-Bellinghausen Sea Low (ABSL) appears as perhaps the most dominant control on indi-

- 25 vidual basin-scale accumulation precipitation variability and associated regionally countervailing accumulation countervailing precipitation patterns. Both the strength and longitudinal location of this low pressure center plays a strong role in determining the center and strength of the dipole in countervailing patterns, which is qualitatively consistent with Hosking et al. (2013). Anomalous deepening of this low (represented as negative pressure changes in z500 plotsvalues, such as the pattern emerging from the basin 26-based composite, Supplementary Material) drives increased snowfall precipitation in eastern WAIS (due to
- 30 enhanced onshore maritime flow) and decreased snowfall precipitation in western WAIS (due to the increased occurrence of offshore continental flow). Conversely, weakening of the ABSL promotes an opposite response (e.g., as represented by the basin 20 composite). Additionally, the longitudinal position of the ABSL plays an important role in determining the position and strength of countervailing dipole in snowfall variability. For example, when the ABSL is shifted to the east (as represented by the basin 22 composite) the dipole is weakened because the resulting pattern of circulation is characterized by strong southerly

flow over the Weddell Sea that precludes the large increases to continental outflow over the AIS that would compensate for increased snowfall to the west. Conversely, a westerly shift in the center of variability(e.g. the basin 20/26-based composites) displays a much higher countervailing pattern in basin-integrated snowfall patterns because in this case the opposite arm of the circulation center that exhibits southerly flow does extend over the Antarctic land mass. Our findings of a dominant ABSL

5 center of variability and regionally counteracting responses to ABSL change is fully consistent with a reanalysis-based analysis (Genthon et al., 2003) that links this variability to Southern Annular Mode and low-latitude Pacific (ENSO) variability.

While ABSL stands out as a dominant center of variability, it also is closely linked and often embedded within broader modes of atmospheric circulation variability that also exert broader exert additional controls on Antarctic snowfall-precipitation patterns. Basin-scale compositing of several East Antarctic drainage basins, most prominently in the vicinity of Enderby Land

- 10 and Dronning Maud Land, reconstruct a notable wave-3-like difference pattern in z500. A relatively more subtle wave-three pattern also emerges in the basin-scale compositing of several AP basins and also basin 14 (George V Land) hinting at the presence of sensitivity to wave-3 variability at nodes separated by approximately 120° around Antarctica. This is consistent with the presence supports a suggestion of a quasi-stationary, temporally variable wave-3 circulation in CESM that impacts Antarctic snowfall precipitation along southward-flowing branches of the wave train. The same pattern, at least in the Enderby
- 15 Land/Dronning Maud Land region, also is associated with coherent remote changes in z500 and moisture transport, leading to AIS-wide patterns of both muted (but significant) positive and negative inter-basin correlations, highlighting non-annularity in wave-three circumpolar circulation as an important control on countervailing inter-basin snowfall variability patterns. This finding is strongly supported by previous studies which. Suggestions of a wave-3-driven signature on Antarctic precipitation aligns with previous studies that identify the effect of wave-3 circumpolar circulation non-annularity of zonal atmospheric
- 20 circulation (e.g. as quantified by the ZW3 indice, Raphael, 2004), with increasing (e.g. as quantified by the ZW3 index, Raphael, 2004) on meridional moisture transports onto the AIS. However, it contrasts with an assessment that wave-3 non-annularity closely associated with increasing meridional moisture and heat transports . structures do not contribute significantly to Antarctic precipitation variability (Genthon et al., 2003). This discrepancy perhaps arises because the signature of wave-3 variability on Antarctic precipitation is swamped in the Antarctic-wide EOF analysis of the latter study, which is dominated by circulation
- 25 variability in the West Antarctic sector.

For a further set of basins, basin-scale snowfall variability appears most linked to local asymmetries in the broad zonal-wave 1 atmospheric circulation. In some cases (e.g. basin 10), basins tends tend to receive more snowfall precipitation primarily as a result of a regional dipole of increased continental and decreased offshore z500 change. This pattern suggests, pointing to the presence of a re-occurring regional elongation of zonal wave 1 asymmetry in circumpolar circulation, that is weakly reflected in

30 more remote circulation changes. In other cases (e.g. basin 17, encompassing the South Pole), increased snowfall precipitation appears associated with a broader pattern of higher continental and lower maritime z500 height wave 1 variability that is circumpolar in nature, and is. It is therefore qualitatively similar to the pattern of Southern Annular Mode (SAM) variability that is highlighted as a strong control on precipitation variability in AIS-wide principal component analysis (Genthon et al., 2003).

**4 Discussion**

In this study we identify the presence of significant dampening and actively countervailing tendencies in CESM-simulated Antarctic annually-averaged snowfall precipitation variability, the dominant term in the Antarctic surface mass balance. Our finding of countervailing Antarctic snowfall variability patterns in CESM agrees with a finding of similar patterns in RACMO2

5 and also in ice-core-based assessments(Thomas et al., 2017). These similarities give confidence that CESM-simulated patterns of AIS snowfall-precipitation variability are physically realistic.

Counteracting patterns of basin-integrated CESM-simulated Antarctic snowfall precipitation variability are directly linked to - and driven by - variability in broad-scale atmospheric circulation patterns, which impact atmospheric moisture transport pathways. Thus, an important finding of this study is that an intrinsic characteristic of Antarctic atmospherically-controlled

- 10 snowfall patterns over Antarctica is suppression of ice-sheet-integrated variability by opposing precipitation variations. Our compositing approach The compositing approach used here isolates spatial patterns of atmospheric variability that promote dampening/counteracting of the impact of variability for each ice sheet drainage basinthis effect, via simultaneous changes to other basins. These basin-specific composite patterns closely resemble known modes of variability that are well known regulators of Southern Hemispheric climate. These include, for example, Amundsen Sea Low ABSL variability, the Southern
- 15 Annular Mode, SAM and wave-3 non-annularity in zonal atmospheric circulation. In reality, these modes of variability are superimposed in a complex manner that likely includes inter-mode interactions (e.g. Fogt et al., 2011; Hosking et al., 2013). Yet, it is clear from the initial analysis presented here that each mode has a distinct signature on AIS patterns, and the combined signatures act to strongly dampen AIS contribute to a dampening of AIS-wide integrated variability. Previous work has identified spatially counteracting impacts of regional atmospheric circulation variability over WAIS that are focused
- 20 around West Antarctica (Genthon et al., 2003; Genthon and Cosme, 2003; Genthon et al., 2005). The findings we present here improve on these important initial analyses, by extending the description of counteracting signals across the entire ice sheet.

In many cases, basin-specific accumulation precipitation compositing reveals similar patterns of snowfall-precipitation variability for nearby basins, including similar distal inter-basin patterns of countervailing variability-variability relationships. This reflects an expected dependence of nearby basins on similar patterns of regional atmospheric variability. However, in some

- 25 cases, immediately adjacent basins show no statistical correlation despite large moisture transport vector changes (for example, basins 19 and 22, Figure 4d). In these cases, lack of correlation in snowfall variability occurs because the dominant modes of regional atmospheric variability that drive large changes in onshore moisture transport in one basin drive simultaneous changes in moisture advection that are directly cross-shore over the other, resulting in a change in direction in of prevailing moisture transport but insignificant net change in transport magnitude or pathway length.
- 30 In other cases, adjacent basins exhibit not just a lack of correlation, but rather significantly anti-correlated snowfall-variability. The presence of such patterns reflects the location of ice divides coupled to patterns of atmospheric flow anomalies. For example, increased basin 2 snowfall precipitation is correlated with decreased snowfall precipitation in multiple basins on the other side of East Antarctic ice divides, because the coherent large-scale change in atmospheric circulation that brings flow onshore over basin 2 acts simultaneously to increase continental outflow in these other locations. As a result, many Antarctic

ice divides practically mark strong transitions in inter-basin variability phasing. This may be an additional important cause of notably decreased accumulation precipitation CV along these divides (Figure 2c), as they are influenced by synoptic systems representing counteracting modes of variability from either side of the divide over the course of a surface mass balance year.

Use of interbasin correlations calculated over the full 1800 year record clearly defines long term mean inter-basin interannual

- 5 correlation strengths, but masks multi-decadal to multi-centennial variability in these same relationships. To expose this low frequency variability, in a similar manner to Genthon and Cosme (2003) we re-calculated inter-basin correlations (here using a 31-year moving window) to define the period over which inter-basin time series were evaluated for correlation strength. Figure 7 demonstrates the time evolution of three inter-basin correlations associated with basin 19 (western West Antarctica), the basin exhibiting both the highest and lowest long term mean inter-basin correlations: a highly positively correlated time
- 10 series (between basin 19 and neighbouring basin 18); a highly negatively correlated time series (between basin 19 and basin 25, eastern West Antarctica); and in the middle, a weakly correlated time series (between basin 19 and basin 4). All other inter-basin evolving correlation time series are available in the Supplementary Information.

In the positively-correlated basin 19/18 case, the 31-year windowed correlation coefficient varies by  $\sim 0.13$  (peak to trough), with notable variability arising at near-centennial timescales. In comparison the correlation coefficient in the negatively-correlated

- 15 basin 19/25 varies by over 0.5, presumably reflecting strong centennial-scale variability in ABSL conditions. Finally, the correlation coefficient in the weakly-correlated basin 19/4 case varies very strongly (~0.8). In addition and somewhat remarkably, the correlation time series in this last case exhibits century-scale periods of both negative and positive correlations. This indicates for this particular inter-basin relationship the potential for precipitation to vary both in phase and out of phase, with these different regimes able to persist for well over a century.
- 20 The presence of significant low-frequency in inter-basin precipitation correlations (demonstrated for basin 19, and qualitatively present in other inter-basin correlation time series) likely has implications for interpretation of ice core interrelationships and also modelled and observed precipitation patterns over the recent historical period. In the former case, the finding of significant low frequency variability about the long term CESM-simulated mean points to climate models as important tools for mechanistic understanding of poorly-understood low-frequency changes in ice core records (e.g. Fudge et al., 2016). In the
- 25 latter, the simulation of centennial-scale variability in inter-basin relationships suggests caution is warranted when interpreting Antarctic recent modelled/observed precipitation changes, since it likely represents an aliased signal of centennial-scale precipitation variability.

We are confident in our qualitative conclusion that an intrinsic aspect CESM-based interpretations of atmospheric circulation over Antarctica is its ability to actively dampen AIS-integrated snowfall variability controls on inter-basin Antarctic

- 30 precipitation variability relationships. This confidence arises from corraboration corroboration of CESM results by similar RACMO2 results, as well as indications of counteracting snowfall qualitatively similar precipitation patterns from ice core records (Thomas et al., 2017). However, some fraction of the CESM-simulated structure of dampened and countervailing snowfall-inter-basin variability that CESM demonstrates may reflect model bias in both CESM-simulated mean atmospheric circulation variability. For example, Holland et al. (2016) note a general climate model bias to-
- 35 wards too-weak modes of non-annular variability in zonal atmospheric flow (e.g. Raphael, 2004) which, if present in the the

CESMLE preindustrial simulation, would spuriously reduce that is shared by CESM. This CESM bias is likely spuriously reducing basin-scale counteracting variability, thereby dampening the pattern of counteracting basin-scale variability and potentially explaining the relatively lower magnitude of (anti)-correlation values relative to RACMO2 (Figure 3). In a similar vein, the ABSL tends to exhibit notable biases in seasonal longitudinal position across a range of climate models (Hosking

- 5 et al., 2013)that if present in CESM could impact spatial patterns of countervailing variability via control of ABSL location on inter-basin variability correlations. This dynamical bias , that. However, this dynamical bias is notably minimized in earlier simulations of CESM (Hosking et al., 2013), could potentially account for discrepancies in correlation patterns CESM (Hosking et al., 2013; Holland et al., 2016) and is thus unlikly to impact differences between CESM and RACMO-RACMO2 interbasin correlations (Figure 3) although the short RACMO. Finally, we also note that errors in RACMO2-simulated
- 10 inter-basin correlation estimates could arise simply from short RACMO2 time series lengthcould also play an important role. We argue that countervailing Opposing tendencies in CESM-simulated Antarctic snowfall variability have precipitation

variability has a strong impact on integrated annually averaged Antarctic snowfall variability. This is, quantitatively reflected in the CV of CESM-simulated AIS-integrated snowfall-precipitation (0.04) being  $\sim$ 4x smaller than basin-scale averaged CV (0.17). In the counterfactual case where these dampening and countervailing patterns were entirely absent and the Antarctic

- 15 continent varied synchronously with a CV of 0.17, the CESM-simulated AIS integrated snowfall variability would increase by  $\sim$ four-fold to a 1-sigma standard deviation of 416 Gt/yr, or approximately 1.2 mm/yr of global mean sea level (GMSL) variability. This The additional variability would be comparable in magnitude to 20th century secular sea level rise rates ( $\sim$ 1.7mm/yr, Church et al. (2013)). Furthermore, it would add notably to GMSL interannual variability, which is dominated by ENSO-driven land water storage changes that generate up to  $\sim$ 5 mm of sea level anomaly (Llovel et al., 2011)). Thus,
- 20 dynamically-linked dampening/counteracting Antarctic snowfall we argue that atmospherically driven counteracting Antarctic precipitation variability patterns play an important role in moderating GMSL changes by removing a large source of variability from the GMSL budget.

We also Similarly, we suggest that the presence of AIS snowfall precipitation variability dampening/counteracting patterns plays an important role in the detection of forced Antarctic mass change signals, because it reduces the effective noise that

- the signal of forced changes to Antarctic snowfall must overcome to become statistically detectable. For example, Previdi and Polvani (2016) highlight the potential of interannual Antarctic snowfall precipitation variability to mask a forced signal of snowfall-precipitation increases over the 1961-2005 period. Our results suggest that the already-strong masking of forced trends by interannual variability would be even more effective in the case where dynamically-driven dampening/counteracting patterns in Antarctic snowfall-precipitation were absent. Similarly, Wouters et al. (2013) highlight the role of Antarctic mass
- 30 balance variability in obscuring assessments of ice sheet mass loss acceleration over the recent observational timeframetime frame. Our results indicate that in the absence of dampened/countervailing Antarctic snowfall opposing Antarctic precipitation variability patterns, this already significant impediment to assessing integrated Antarctic mass changes would be much greater.

**5 Conclusions**

10

In this study we analyze climate model output from the Community Earth System Model Large Ensemble preindustrial control simulation to explore patterns of snowfall precipitation variability over Antarctica. Most notably, correlation analyses of CESM-simulated snowfall integrated over precipitation between Antarctic ice drainage basins highlight the presence of

5 highlights strong patterns of dampened and even countervailing opposing basin-integrated snowfall precipitation variability. The physical veracity of this finding is strongly supported by the presence supported by very similar behaviour by the regional RACMO model, and further confirmed by indications of countervailing precipitation in regional modeling, global reanalysis products and ice-core records.

Compositing of climate conditions based on the annual time series of snowfall annual precipitation in each basin permits investigation of the broader climate conditions associated with basin-scale integrated snowfall precipitation variability. This

- analysis reconstructs previously-characterized known modes of Southern Hemisphere high latitude atmospheric circulation variability, confirming these modes as dominant controllers of both basin-scale snowfall variability and inter-basin countervailing variability patterns. This The control arises because of variability in atmospheric moisture transport, which is closely linked to regionally countervailing trends in Antarctic snowfall strongly regulates Antarctic precipitation patterns and also drives re-
- 15 gional sea ice variability. Important modes Apparently important sources of variability with respect to regionally countervailing patterns of Antarctic snowfall precipitation include the Amundsen/Bellingshausen Sea Low and the zonal wave three pattern in circumpolar atmospheric circulation, and variability of the Southern Annular Mode. The presence of simulated decadal and centennial scale variability in inter-basin correlations highlights the potential for low frequency shifts in Antarctic precipitation variability regimes. Finally, we suggest that the presence of widespread dampening of snowfall variability likely dampening of
- 20 integrated Antarctic precipitation variability due to opposing inter-basin variability patterns has a notable impact on integrated Antarctic ice sheet Antarctic mass variability, with important potential implications for the detection of ice sheet mass change accelerations, emergence of an anthropogenically forced signal in Antarctic integrated snowfall, precipitation and global sea level rise changes.

*Code and data availability.* All NCAR Command Language (NCL Version 6.4.0, 2017, http://dx.doi.org/10.5065/D6WD3XH5) analysis
 scripts are available at: github.com/JeremyFyke/AIS\_snowfall\_analysis. Directions for obtaining CESM Large Ensemble data are available at: www.cesm.ucar.edu/projects/community-projects/LENS/.

Competing interests. The authors declare no competing interests.

Acknowledgements. We thank Christophe Genthon, Hansi Singh, Marika Holland and Nicole Jeffrey for useful feedback. J. Fyke and H. Wang are supported in this work by the HiLAT project, funded by the US Department of Energy Regional and Global Climate Modeling program. Jan Lenaerts is supported by the Netherlands Science Organization through the Innovational Research Incentives Scheme Veni. We thank the Community Earth System Model Large Ensemble coordinators for making simulation output available.